# Epidemiological Estimate of Growth Reduction by Ozone in *Fagus sylvatica* L. and *Picea abies* Karst.: Sensitivity Analysis and Comparison with Experimental Results

**DOI:** 10.3390/plants11060777

**Published:** 2022-03-15

**Authors:** Sabine Braun, Beat Rihm, Christian Schindler

**Affiliations:** 1Institute for Applied Plant Biology AG, CH-4108 Witterswil, Switzerland; 2Meteotest AG, CH-3014 Berne, Switzerland; beat.rihm@meteotest.ch; 3Swiss Tropical and Public Health Institute, University of Basel, CH-4002 Basel, Switzerland; christian.schindler@unibas.ch

**Keywords:** ozone flux, dose–response, epidemiology, sensitivity analysis, *Fagus sylvatica*, *Picea abies*

## Abstract

The critical level of ozone flux for forest trees is based entirely on biomass data from fumigation experiments with saplings, mostly in open-top chambers. Extrapolation to mature forests asks, therefore, for validation, which may be performed by epidemiological data analysis. This requires a multivariable regression analysis with a number of covariates to account for potential confounding factors. The present paper analyses the ozone sensitivity of volume increments of mature European beech (*Fagus sylvatica*) and Norway spruce (*Picea abies*), with the addition, or removal, of covariates. The comparison of the epidemiological dose–response relationship with experimental data shows very good agreement in beech and a more sensitive relationship in the epidemiological analysis of Norway spruce compared to the experiments. In Norway spruce, there was also a strong interaction between the effects of ozone and temperature; at high July temperatures, the ozone effect was stronger. This interaction may explain the disagreement between the epidemiological study and the experiments, of which the majority were performed in Sweden.

## 1. Introduction

The critical level for tropospheric ozone is based on the dose–response relationships with biomass data from fumigation experiments with young trees in chambers [1]. A set of fumigation experiments, mostly in open-top chambers (OTC), was compiled, and the biomass response at a specific ozone flux was related to the biomass of the controls. Ozone flux, i.e., the stomatal uptake of ozone during a season, proved to be a better predictor than the accumulated ozone over a threshold of 40 ppb (AOT40, [2]). Dose–response relationships were established for beech/birch and for Norway spruce [3], and critical levels were set to 5.2 (4% biomass reduction for beech and birch) and 9.2 mmol m^−2^ yr^−1^ (2% biomass reduction for Norway spruce) [4].

The drawbacks of these kinds of experiments are the use of young trees and a chamber effect on climate [5]. A more realistic estimate can be obtained by either free-air fumigation of larger trees [6,7] or by epidemiological analysis of forest data [8]. The latter was used by Karlsson [9], who showed a negative impact of ozone, as AOT40, on the stem basal area increment of mature Norway spruce in South Sweden. Epidemiological analysis has also been applied by Sicard [10], to derive a critical level for visible ozone injury. Other authors did not find an ozone effect. Paoletti [11] did not find an ozone effect on stand volume growth in 728 European beech sites across Italy in one five-year increment period. Verryckt [12] looked for short-term ozone effects on gross primary productivity in one Scots pine stand, and also did not find an effect. These somewhat contradictory results raised doubts about the validity of epidemiological methods [13]. The aims of this paper are, therefore, to take a closer look at the potential reasons for this contradiction, as well as the following:To present a sensitivity analysis on the covariates for the epidemiological analysis of ozone effects on beech and Norway spruce.To compare observational data on adult trees with the experimental dose–response functions, using volume increment as the dependent variable. Volume increment allows a direct comparison to be made with the experiments, which are based on biomass.

## 2. Materials and Methods

### 2.1. Plots

The network of the intercantonal forest observation programme is described in [14]. Beech was monitored in 91 plots, and Norway spruce in 72 plots, for up to 32 years. Each plot consists of 60–70 mature trees. Total number of trees available in the current analysis was 7427 beech and 6538 Norway spruce. The plots cover (a.o.) gradients in age, drought, altitude, N deposition, soil base saturation and ozone flux, thus allowing a multivariate statistical data analysis to be performed. Leaf chemistry was analysed every four years, and soil chemistry every twelve years.

### 2.2. Tree Increment

Diameter at breast height was measured every four years on marked points in all trees of a plot. In a subset of trees per plot, tree height and diameter at a height of 7 m was measured. Tree volume was calculated using the formula given in [15]. The formula had the form b_0_ + b_1_ × term_1_ + b_2_ × term_2_… + b_n_ × term_n_, with the coefficients listed in Table 1. Volume increment was obtained by calculating the volume of trees at the beginning and at the end of each increment period. The data set consists of 8 increment periods.

### 2.3. Ozone Flux

Ozone flux was calculated for 30 rural monitoring stations in Switzerland for the years 1991–2019 using the model DO3SE [16]. The annual sums were mapped as described in [17], and the map estimates per site and year were used in the current analysis. Figure 1 shows the average ozone flux for beech for the last increment period, 2015–2019, together with the location of the beech plots used for this analysis.

### 2.4. Data Analysis

Volume increment was used as the dependent variable. It was square root transformed and analysed in a linear mixed regression model, with site, year and tree as the random variables. The function glmer in R was used [18]. The initial set of predictors was selected based on theoretical considerations. It included site factors (age and stand density), tree factors (social position, position within stand, crown size, initial diameter at breast height (DBH), foliar nutrient concentrations, soil chemistry, nitrogen deposition, ozone, air temperature and drought. Potential problems with collinearity between the predictors were checked by calculation of the variance inflating factor (VIF, [19]). All covariates were centred and scaled at their mean before regression analysis. Variables increasing the Akaike information criterion (AIC) were removed, and effects were examined for biological plausibility. After selection of the main effects, nonlinear effects were tested using the function poly. When this introduction decreased the AIC, the nonlinear response was maintained. After this, all possible pairwise interactions were introduced into the model and the interactions that increased the Bayes information criterion (BIC) were removed. This latter criterion is more restrictive than the AIC and helps to reduce the number of degrees of freedom of the final model. Residuals were checked visually for normal distribution and outliers. They were plotted against the predictors as well as against time (also without the inclusion of year as a cluster). The results table was produced using the function “Anova” (R library car, [20]). In the table, the polynomial degree in nonlinear relations is indicated in the column “poly”. 

Graphical representations of the regression results were made using the R function ggpredict [21], which sets all other variables to their mean and the random effects to zero. To illustrate interactions, predictions were extracted for two levels of the second predictor. 

The following covariates were included in the model:Annual ozone flux (POD_1_); see Section 2.3.Nitrogen deposition: mapped total deposition in 1 ha spatial resolution, see [22].Drought: Various drought indicators were tested [23]. The best drought indicator for beech was the ratio between actual and potential evapotranspiration, averaged between 5 days before and 80 days after budbreak. For Norway spruce, it was the minimum site water balance per year.Age: Stand age in years (baseline at the start of the time series). This was kept constant during the time series, as recommended for cohort studies [24].Stand density index according to [25].Tree diameter at the beginning of the increment period.Crown size (crown projection area).Social position (dominant, codominant or suppressed) and position within the stand (stand and edge/gap).Fructification: number of fruits per short shoot (beech), analysed retrospectively from shoots collected for foliar analysis.Proportion of deciduous trees in the stand.Foliar concentrations of Ca, P and K [26].

The following sensitivity analyses were performed:Dependency of the ozone effect on the ozone model grid size (beech data set). Ozone was calculated as modelled (grid size 0.25 km), and averaged over grid sizes of 1, 5, 10 and 50 km.Dependency of the ozone effect on the inclusion and on the removal of covariates. The effect of covariates on the ozone effect was tested both backwards (removal of covariates from the full model) and forwards (addition of covariates to a model with ozone only).Comparison of the slopes with dose–response relationships from OTC experiments [27] and from the free-air fumigation in Kranzberg [28]. To enable a direct comparison of the experimental results with the results from epidemiology, curves were aligned on a scale of relative biomass with 100% corresponding to PODi = 0. The two points from the Kranzberg experiment also relate to a relative biomass of 100% at POD_1_ = 0. This has been the standard procedure for establishing the dose–response curve from the experiments [29].

## 3. Results

### 3.1. Model Results for Beech and Norway Spruce

Table 2 shows the regression results for both beech and Norway spruce. Interactive terms, which neither refer to ozone nor to nitrogen deposition, are omitted from this table for reasons of clarity. In beech, there were 13 interactions, and in Norway spruce, there were 15 interactions.

Ozone is clearly negatively related with volume increment in both beech and Norway spruce. Ozone significantly improves the regression (Table 3). In beech, the ozone response was stronger when fructification was high (not shown), and in Norway spruce, when the temperatures in July were high (Section 3.3). In both species, there was an interaction between ozone and nitrogen deposition, but the direction of the interaction differed between species; in Norway spruce, the ozone response was weaker at high N deposition, whereas in beech, it was stronger (Figure 2).

### 3.2. Effect of Grid Size on the Ozone Model for Beech

The grid size of the ozone model does not affect the ozone regression result very much, although the confidence interval slightly increases with larger grid sizes (Figure 3). 

### 3.3. Effect of Covariates on the Regression Result

The removal of covariates in beech allows us to identify the relevant covariates and to identify the ozone effects. Not considering fructification and N deposition had the largest impact on the estimated ozone effect in beech. These two predictors also interact with ozone. It was, therefore, important to include respective interaction terms. The estimated effects of ozone flux also decreased after the removal of drought (Figure 4). In Norway spruce, the removal of temperature had the largest effect, followed by the removal of interactions and of N deposition. In this case, too, the predictors with significant interaction terms caused the largest effects. The removal of stand density increased the ozone effect.

The removal of all covariates results in much weaker ozone effects; in beech, the effect is no longer significant (Figure 4; “all covariates removed”; Figure 5; “no covariates”). While adding drought as the first covariate has almost no effect on the ozone effect, the addition of N deposition results in a clearer increase. In Norway spruce, the ozone effect without covariates is already significant, and the coefficient is about half of that in the full model. The inclusion of drought resulted in an ozone effect similar to the full model, while the inclusion of temperature, as the only additional variable, resulted in complete loss of the ozone effect. The latter is the result of the strong temperature interaction. 

### 3.4. Comparison with Dose–Response Relationships

In beech, the dose–response curve from the epidemiological model corresponds with the dose–response curve from the OTC experiments, and also with the results from Kranzberg free-air fumigation (Figure 6). In Norway spruce, the epidemiological dose–response curve is much steeper than the dose–response curve from the experiments. There was a highly significant interaction between ozone and maximum July temperature (Figure 7), which suggests that in colder areas, the ozone effect is smaller than in warmer areas.

## 4. Discussion

The current analysis shows a clear ozone effect on volume increment for both beech and Norway spruce, but also the importance of considering appropriate covariates in a multivariate analysis. In the study of [11], the statistical methods are not described in sufficient detail, and the written interpretation contradicts the only results table that displays the Pearson correlation coefficients with a significant negative correlation between growth and drought (soil water content) and POD_0_. Moreover, the study is based on only one five-year period, without intermediate measurements. In the multivariate correlation analysis, N deposition was probably only partly represented, as the correlation table only shows two of its components (NO_2_ and NH_3_). The size of the data set must also be large enough to be able to show the effects, which may be the reason for the non-significant result in [12]. In addition, gross primary production, as analysed in [12], is not the same as growth [30], which is the basis of the dose–response relationships used for setting the critical level [3]. In any case, a non-significant regression result cannot be interpreted as the absence of any effect.

The sensitivity analysis for model grid size suggests that even coarse grids can be used for the epidemiological analysis of ozone effects on trees, and that the temporal variation between years is more important than the spatial variation between plots. This is in contrast to a similar analysis, which was conducted by Kohli (cited in [8]), for nitrogen deposition, where the effect of N deposition on vegetation changes was no longer statistically significant when using larger grids. The importance of temporal variation implies that the analysed data cover a sufficiently long time period, and include a sufficient number of repeated measurements. The current data set covers eight incremental periods, each of 4 years, i.e., a total of 32 years.

Our estimated effects of ozone flux for beech were more sensitive to the presence of other predictors than the estimated effects for Norway spruce, as the model without covariates was not significant in beech, but was in Norway spruce. The most important predictors to include in the beech model were N deposition and fructification; in the spruce model, the most important predictors were N deposition and temperature. These were the variables that showed significant interactions with ozone. It is also important to have a good drought indicator. Precipitation alone is not sufficient [23]. 

N deposition modified the ozone response. This modification differed between beech and Norway spruce. In beech, N deposition slightly increased the ozone sensitivity, which might be explained by increased stomatal conductance and, thus, ozone uptake [31,32]. The ameliorating effect of N deposition on the ozone response, as observed in Norway spruce, is observed more often, but both antagonistic and synergistic responses have been described [33,34].

The agreement of the dose–response curve from our epidemiological model with the experimental data is good in beech, and suggests that the current critical level for ozone is also reliable for mature forests. The disagreement in Norway spruce may be a consequence of the observed temperature interaction. Twenty-one points in the experimental dose–response curve stem from a fumigation experiment in Östad, Sweden, six from a Swiss site at 1000 m altitude (Zugerberg), and only two from a Swiss site at low altitude, in a temperate zone (Schönenbuch). 

These comparisons between the experimental and epidemiological results suggest that the critical level for ozone in Norway spruce may be too high, and that the losses in the Swiss forests, due to ozone [17], may be underestimated in coniferous forests.

## 5. Conclusions

The current data analysis presents recommendations for estimating the effects of ozone on biomass increments of mature forests using epidemiological data. Firstly, measurements of important covariates that affect growth are required (in the current data set, these are drought, temperature, N deposition, forest structure, and stand age). These covariates must be included in a multivariate model. Secondly, the time series of ozone flux must be long enough to include sufficient temporal variation in ozone, and the number of plots should be higher than 10 times the number of site-level covariates [8]. The very good agreement between the epidemiological data analysis and experimental data in beech suggests, on the one hand, that the experiments are not confounded by the use of young trees and, on the other hand, that an epidemiological data analysis is able to provide unbiased estimates of the ozone effect in mature beech. To find an explanation for the disagreement between the epidemiological and experimental dose–response relationships in Norway spruce, a larger data set, with broader geographical coverage, might be useful.

## Figures and Tables

**Figure 1 plants-11-00777-f001:**
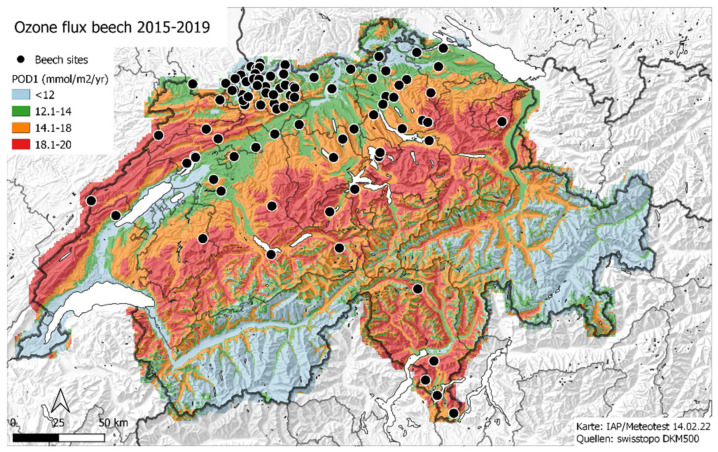
Map of beech ozone flux (average 2015–2019) and location of the beech plots used for the study.

**Figure 2 plants-11-00777-f002:**
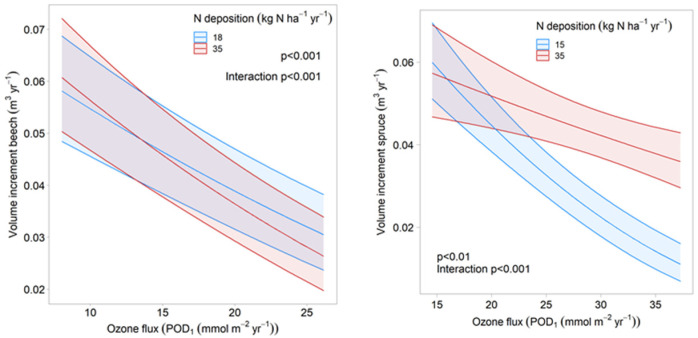
Interaction between ozone and nitrogen deposition in beech (**left**) and Norway spruce (**right**). Ozone response extracted from the regression model for two levels of nitrogen deposition using ggpredict (i.e., by setting all other variables of the model to their mean values).

**Figure 3 plants-11-00777-f003:**
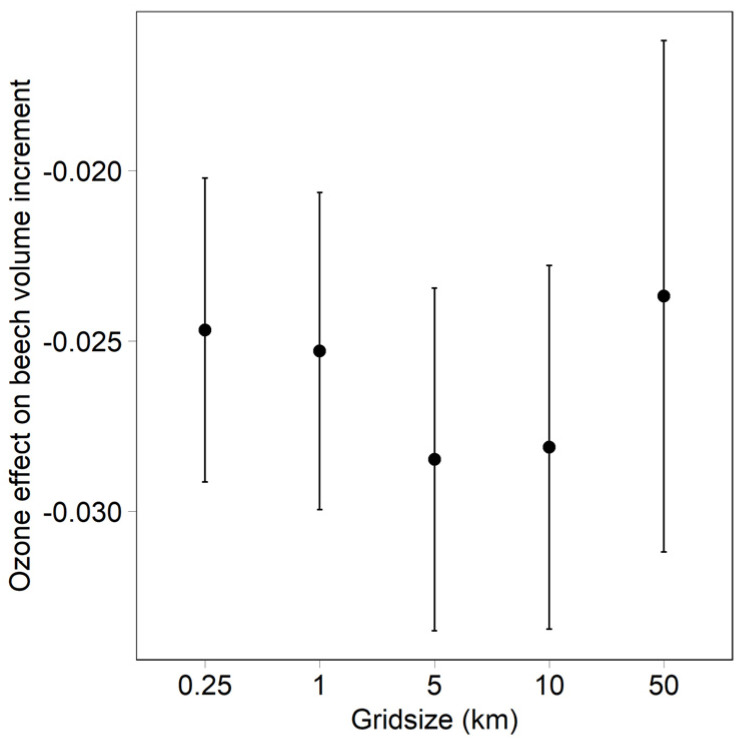
Effect of the grid size of ozone model on the ozone coefficient in the regression. Bars indicate 95% confidence interval.

**Figure 4 plants-11-00777-f004:**
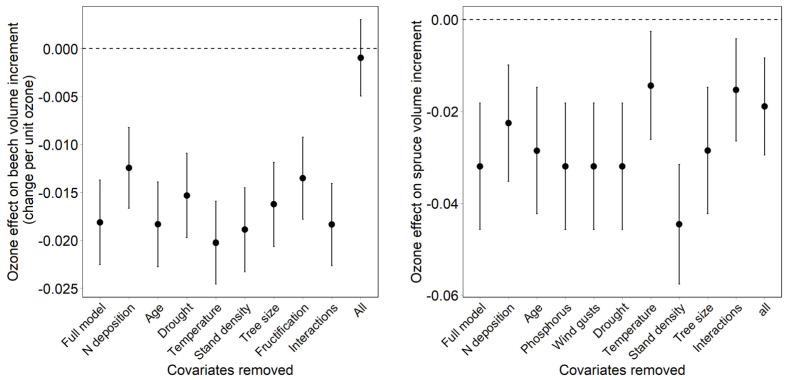
Effect of removal of covariates on the ozone effect in comparison to the full model for European beech (**left**) and Norway spruce (**right**). All = all covariates removed. Bars = 95% confidence interval.

**Figure 5 plants-11-00777-f005:**
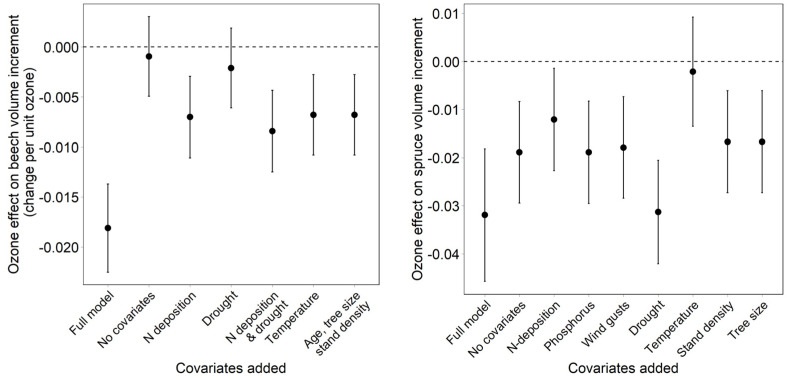
Changes in the ozone effect after adding single covariates to the model with ozone only (“no covariates”) and after adding all covariates (“full model”). (**Left**): European beech; (**right**): Norway spruce.

**Figure 6 plants-11-00777-f006:**
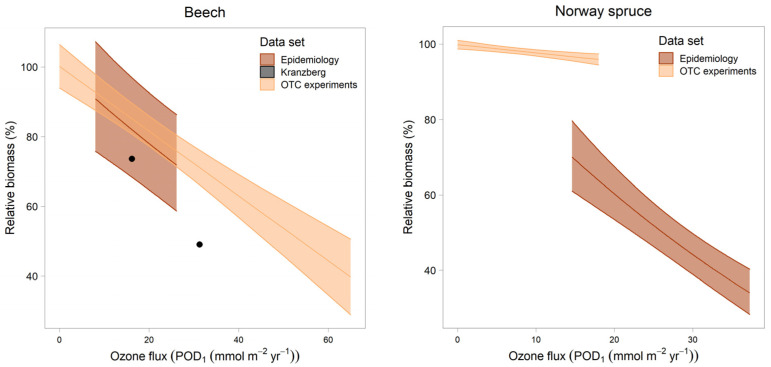
Dose–response relationship from the regression model described in this paper (“Epidemiology”) in comparison to the dose–response curve from experiments (“OTC experiments”; [27]). The results from both the epidemiological analysis and the experiments were adjusted to reach 100% at POD1 = 0. In the case of beech, the results from Kranzberg free-air fumigation (“Kranzberg”) are shown as black dots. Dose–response curves for the epidemiological data were obtained using ggpredict (i.e., by setting all covariates to their mean values).

**Figure 7 plants-11-00777-f007:**
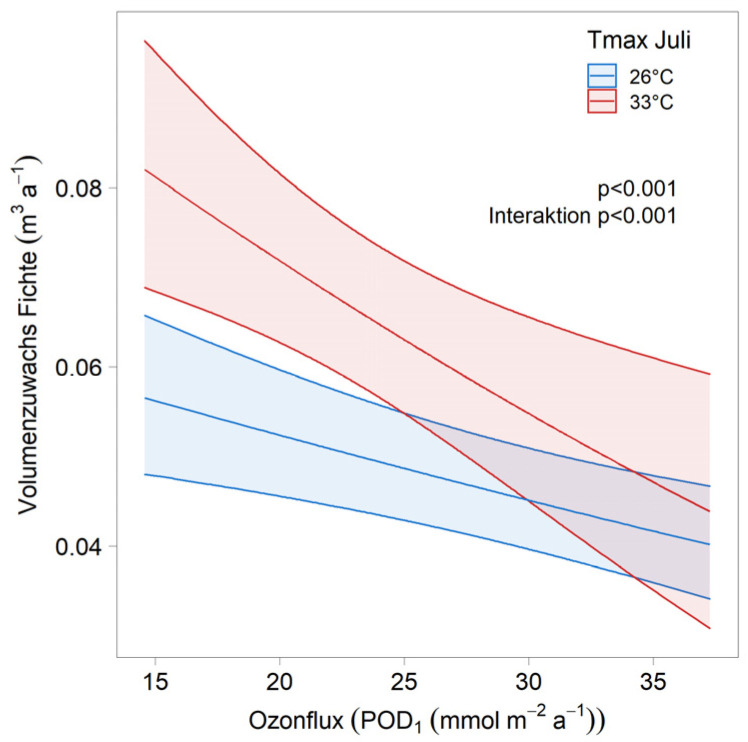
Interaction between ozone and maximum July temperature in Norway spruce. Dose–response curves for different levels of maximum July temperatures (i.e., 26° vs. 33°) were obtained by including an interaction term between ozone flux and maximum July temperature. Dose–response curves were obtained using the function ggpredict for varying levels of ozone and two fixed levels of temperature, while setting all other covariates to their mean values.

**Table 1 plants-11-00777-t001:** Coefficients for the calculation of tree volume [15].

	*Fagus sylvatica*	*Picea abies*
term	coefficient	coefficient
b_0_	24.732	5.8202
H	−1.8734	
H × d	0.18617	
D × H × d	0.037304	
D × d^2^	0.0078491	−0.0077515
D^2^ × H × d	−0.00021339	
D × d		0.16032
D^2^		0.26975
d^2^		−0.2776
D^2^ × H		−0.0054489
H × d^2^		0.052114
D^2^ × d		0.0046137

D: diameter at breast height (DBH, cm); d: diameter at 7 m height (cm); H: shaft length (m).

**Table 2 plants-11-00777-t002:** ANOVA table of the regression results for European beech and Norway spruce. Poly: number of polynomial degrees in the regression. Chisq: Χ^2^ from the ANOVA test. All entries are highly significant (*p* < 0.001). Only interactions involving ozone, nitrogen deposition or drought are shown.

Response: Volume Increment	Level	*Fagus sylvatica*		*Picea abies*	
		Chisq	Poly.	Chisq	Poly.
Number of trees		7427		6538	
Number of plots		91		72	
Number of observations		32,907		23,669	
R^2^ marginal		0.360		0.523	
Age	stand	20	1	19	1
Social position	tree	36	1	73	1
Position within stand	tree	220	1	38	1
Diameter at start	tree×year	1906	2	1040	1
Crown projection	tree	133	2	194	2
Relative stand density	Stand × year	14	1	4	2
Proportion deciduous	stand	34	1		
Fructification	Stand × year	34	1		
Foliar Ca	Stand × year	1	1	33	1
Foliar N:K	stand × year	36	1		
Foliar P	stand × year			75	2
Drought	stand × year	170	1	130	2
Temperature	stand × year	57	2	44	2
Wind gusts	stand × year			46	1
N deposition	stand × year	117	2	128	2
Ozone	stand × year	49	1	12	1
N deposition × drought	stand × year			68	2 × 2
Ozone × N deposition	stand × year	57	1 × 2	52	1 × 2
N deposition × fructification	stand × year	56	2 × 1		
N deposition × Tmean May	stand × year	41	2 × 2	54	2 × 2
N deposition × stand density	stand × year			112	2 × 1
N deposiiton × diameter at start	stand × year			72	2 × 1
Ozone × stand density	stand × year	58	1 × 1	44	1 × 2
Ozone social position	stand × year	43	1 × 1		
Ozone × fructification	stand × year	39	1 × 1		
Ozone × Tmax July	stand × year			116	1 × 2
Drought × foliar N:K	stand × year	110	1 × 1		

**Table 3 plants-11-00777-t003:** ANOVA table of the comparison of models with and without ozone variables.

	*Fagus sylvatica*		*Picea abies*	
n	without	with	without	with
par	50	56	59	66
AIC	−103,509	−103,718	−66,761	−66,922
BIC	−103,089	−103,247	−66,285	−66,389
logLik	51,805	51,915	33,439	33,527
deviance	−103,609	−103,830	−66,879	−67,054
Chisq		220.57		175.19
Df		6		7
Pr (>Chisq)	<2.2 × 10^16^ ***		<2.2 × 10^16^ ***	

Values are means ± SD (*n* = 3). *** *p* < 0.001.

## Data Availability

The study did not report any data.

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
