# Peer review of "Epidemiological Estimate of Growth Reduction by Ozone in Fagus sylvatica L. and Picea abies Karst.: Sensitivity Analysis and Comparison with Experimental Results"

_plants, 2022, doi:10.3390/plants11060777_

Round 1
Reviewer 1 Report
Dear authors:
I went through your improved version of the manuscript, and now I think this is suitable for publication.
Thank you very much for answering my questions, and for the thorough revision of your manuscript.
Reviewer 2 Report
Indeed, the article “Epidemiological estimate of growth reduction by ozone in Fagus sylvatica L. and Picea abies Karst .: sensitivity analysis and comparison with experimental results ” is presented in an improved form and the authors clarified the aspects requested.
This manuscript is a resubmission of an earlier submission. The following is a list of the peer review reports and author responses from that submission.
Round 1
Reviewer 1 Report
Dear Editor, dear authos,
These are my comments on the manuscript entitled “Epidemiological estimate of growth reduction by ozone in Fagus sylvatica L. and Picea abies Karst.: sensitivity analysis and comparison with experimental results”. In this paper the authors aim to investigate the limitation of tree growth at high ozone atmospheric concentrations for two separate tree speces: European beach and Norway spruce. The research topic is timely and important, and the authors have gathered a nice dataset. However, I shall rise severe concerns about the overall scientific soundness of the author’s approach. Please, find my detailed response below, divided between major and minor comments:
Major comments:
I have a major concern regarding the statistical approach, as well as the way that this statistical approach is presented to the readers, and further, the way the authors discuss their results. I would have appreciated some measure not only about the significance of the different variables (table 1), but also the overall model performance description (e.g. R2 metric, Root Mean Square Error metric, etc.). That said, I also found that all models tested by the authors assume an ozone effect, but they have not rejected the Null model (i.e. all covariables removed from the model AND ALSO the ozone effect removed from the model), so their statement that there is a distinct effect of the ozone regarding wood volume growth is not supported by their analysis. Does the parsimony of the model improve when considering the ozone effect with respect to the Null model? This has not been answered in the manuscript and it needs to be addressed.
I also have yet another issue with the statistical treatment of the data: it is quite unclear to me how the authors have calculated the slopes in the Figure 5, either from the same model accounting for Temperature, or either by developing two separated dose-response curves for the different levels of Temperature. This is somewhat also extended to Figure 4, in which the authors do not state how the projections of the model were performed (e.g. if this is a sensitivity analysis on the POD effect projected by their model, or it is just the model output when run with the input variables). These are important points to make clear in order to understand the message the authors want to give regarding their research, and I think they have not been fully developed in the manuscript.
Also related: Why the observations by Krankzberg experiment have been (sic.) “vertically moved to lie in the regression curve (…)” [L 169:170]? By doing so (that is, by modifying the amount of growth per unit of ozone concentration increment) you are modifying the data from the previous experiment (you implicitly assume that there is a tree size / age effect by including these variables as covariates in your analysis), so the output from your model is compared against a tuned dataset. It is also extended to the OTC experiments: in Figure 4, at first glance it seems that your model output matches entirely with the experimental model observations… until one realizes that the authors forced the OTC experiments model to match their observations (by changing the intercept? the slope? It is not stated in the document) [L: 166:167]. Therefore, the figures the authors provide and upon which they base all their discussion are comparisons of their model outputs against two datasets that have been strongly modified, and therefore are not reliable enough to reach any of the strong conclusions the authors reach e.g. L 207:208; L 210:217.
Finally, the authors suggest in their discussion that their results do not support the hypothesis of a temperature impact on the interaction between the growth of Norway spruce and ozone atmospheric concentration… which is quite surprising, given that their model is most likely strongly dependent on the Temperature (Figure 2, right Norway spruce panel), which is the variable that is affecting the most the ozone effect on tree volumetric growth.
So, the presentation of the results must be improved. For instance, some maps about stand distribution and ozone concentration are highly recommended to frame your research. Also, the model must be better explained to understand to what extent it fits the data, and to what extent the observed trends are robust. It is worth considering including the original datapoints in Figures 4 and 5, in order to have a visual representation of their distribution. Last but not least, the data from other studies presented in Figure 4 must be the original without any modification on it. If the authors still want to compare their results with results from previous studies (which is commendable and it is a fair point the authors make in their manuscript), maybe they could provide standardized response values (e.g. the % of change in volumetric increment with respect to a reference ozone value, or look at the slope between the volumetric increment per unit of ozone concentration increment).
Minor comments:
Abstract:
- The authors should clarify what are critical for the ozone flux levels (e.g. tree growth, volume increase, etc…)
Introduction:
- Same than in the abstract [L25]
- Verryckt et al. did not fail to show an effect, but instead they did not find any effect. Your statement is not elegant, and it assumes that there is a correlation which somehow Verryckt et al. failed to capture. [L 35:36]
Objectives: Please, rethink the objectives of the paper, as the stated are too general and simple for a research. For instance, objective 3 lacks for a further explanation regarding the importance of the comparison of the natural conditions VS the experimental conditions.
Materials and Methods:
- Check that the sub-section number are correct (currently identified all as “2.1”).
- The “Plots” section is lackluster. It requires a bit more extension and explanation. Also, the authors could really use a map representing the distribution of the stands to highlight the extension of their dataset.
- The formula for the tree volume increment calculation must be provided, as it is a key variable in your manuscript. [L:54]
- [L 109:116] This entire paragraph should be re-written, including a justification (if there is any) for the modification of the models resulting from the experimental datasets, and the scientific soundness of such a modification of the data.
Results:
- [L 119-123] These sound like methods to me…
- Figures: Please clarify the footnotes in figure 4 and 5 to define if the entire model was used, and also in figure 5 if the trends for the two temperature factors are derived from the same model or two different models fit for each Temperature factor.
- Please, provide some evaluation of model performance, such as the goodness of fit (R2) or the Root Mean Square Error, in order for the reader to evaluate the model performance in reproducing the observed trends.
Discussion:
- L [210:212]. This sentence does not fit the text at all in the form it is redacted.
- It is not clear to me why you conclude that your results can exclude the effect of temperature, when it is crystal clear in figures 2 and 5 that you can’t, and that there is a highly negative impact on the relationship between growth and ozone concentration at higher temperatures. How can you discard that growth limitation is not entirely driven by high temperatures in your database, and that ozone concentration only has a spurious correlation with high temperatures? To do so, I suggest a further layer of complexity in your model, which is to consider the effect of the temperature by itself, the Null model, and the effect of the ozone by itself. Only if the ozone model is more explanatory than the temperature model (and the Null model), then you can argue that ozone might be more important than temperature for volumetric increment limitation.
Reviewer 2 Report
The article” Epidemiological estimate of growth reduction by ozone in Fagus syl-2 vatica L. and Picea abies Karst.: sensitivity analysis and comparison 3 with experimental results” intends to analyse the ozone sensitivity of volume increment of mature European beech (Fagus 14 sylvatica) and Norway spruce (Picea abies) using different statistical methods.
The data set covers 8 increment periods with 4 years each, a total of 32 years and the total number of trees available in this analysis was 7427 beech and 6538 Norway spruce.
This article is clearly prepared, the information is well presented and can be applied to a similar analysis on a larger area (at least a region). The sensitivity analysis on the covariates has a major role for the epidemiological study of ozone effects on beech and Norway spruce.
This paper has a major main contribution in the literature by presenting clearly and gradually the way in which the determinations and interpretations of the results were performed.
I think it is important to explain the following terms, based on the idea that this article should be accessed by people from different fields and with different backgrounds.
- Row 62 ”DBH”
- Rows 65, 72 ”AIC”
- Row 73 BIC
It would be interesting to include in the article some sentences about OTC experiments [20] without having to read other bibliographical references.